palaeontology, microbiology

fossil bacteria, fossil fungi, chemical gardens, biomorphs, astrobiology

**Author for correspondence:**
Sean McMahon
e-mail: sean.mcmahon@ed.ac.uk

# Earth's earliest and deepest purported fossils may be iron-mineralized chemical gardens

Sean McMahon[1,2]

[1]UK Centre for Astrobiology, School of Physics and Astronomy, University of Edinburgh, James Clerk Maxwell Building, Peter Guthrie Tait Road, Edinburgh EH9 3FD, UK
[2]School of Geosciences, Grant Institute, University of Edinburgh, James Hutton Road, Edinburgh EH9 3FE, UK

SM, 0000-0001-8589-2041

Recognizing fossil microorganisms is essential to the study of life's origin and evolution and to the ongoing search for life on Mars. Purported fossil microbes in ancient rocks include common assemblages of iron-mineral filaments and tubes. Recently, such assemblages have been interpreted to represent Earth's oldest body fossils, Earth's oldest fossil fungi, and Earth's best analogues for fossils that might form in the basaltic Martian subsurface. Many of these putative fossils exhibit hollow circular cross-sections, lifelike (non-crystallographic, constant-thickness, and bifurcate) branching, anastomosis, nestedness within 'sheaths', and other features interpreted as strong evidence for a biological origin, since no abiotic process consistent with the composition of the filaments has been shown to produce these specific lifelike features either in nature or in the laboratory. Here, I show experimentally that abiotic chemical gardening can mimic such purported fossils in both morphology and composition. In particular, chemical gardens meet morphological criteria previously proposed to establish biogenicity, while also producing the precursors to the iron minerals most commonly constitutive of filaments in the rock record. Chemical gardening is likely to occur in nature. Such microstructures should therefore not be assumed to represent fossil microbes without independent corroborating evidence.

## 1. Introduction

Filaments and tubes composed predominantly of nano- and microcrystalline iron (oxyhydr)oxides and iron (alumino)silicates occur as dense assemblages in diverse rocks of all ages, including submarine hydrothermal chert (jasper) beds and veins [1–5]; fractures, veins, vesicles, and amygdales in numerous marine and terrestrial basalts [6–10]; mineralized cavities in limestones [8,11,12]; and the porous oxidation zones of metal ore bodies [7,8]. These microstructures range from 1 to approximately 50 μm in diameter and up to several millimetres in length, and show complex morphological features taken to indicate a high probability that they are mineral-encrusted microorganisms, including strongly curved growth trajectories, circular cross-sections, discrete spore-like swellings, true bifurcate branching and anastomosis (cross-linking or convergence of adjacent branches), hollowness, and nestedness [1,2,4,6–15]. These structures have long been of palaeobiological and astrobiological interest, particularly as evidence to inform the search for ancient subsurface life on Mars [7,8,16–18]. Precambrian examples have recently been presented as candidates for Earth's oldest fossils [1] and Earth's oldest fossil eukaryotes (fungi) [6]. In some instances, the inference of biogenicity is supported by independent, non-morphological evidence, including the presence of carbonaceous matter or even identifiable biopolymers within filaments that also contain iron minerals [19,20]. However, most reported assemblages lack such evidence, prompting suggestions that the original cells migrated out of the mineralized sheaths during life [4], or decayed or oxidized away after death [11,12,15].

It has often been argued that morphological evidence alone is insufficient to establish that candidate fossil microbes are actually biological in origin; some microbe-like structures are already known to be produced by non-biological processes [21–23]. Despite this, the idea that inorganic filamentous microstructures can display uniquely and identifiably biogenic morphology has persisted. Abiotic models for the origin of the iron-mineralized filament assemblages under consideration in this paper have hitherto been widely rejected, being considered unable to explain many of their features. Some filaments [24,25] are uncemented, positive projections into open (originally fluid-filled) space; many others show petrographic relationships interpreted to show that they formed in open space and were only later surrounded by cavity-filling calcite, quartz, or clay. This evidence excludes any formation mechanism involving the local alteration of a pre-existing solid or highly polymerized medium (e.g. ambient inclusion trails [13] or dendritic diffusion through viscous silica gel [26–28]) which could not have been removed without destroying the entombed microstructures [4]. Fibrous crystals (e.g. metal oxides, illite, serpentine, and palygorskite–sepiolite) can also superficially resemble curving filaments [29] but show angular cross-sections and lack non-crystallographic branching, anastomosis, hollowness, or spore-like swellings.

The experimental dissolution of alkaline earth metal salts in viscous, alkaline silica solutions under $CO_2$ produces a range of filamentous carbonate and silica-carbonate 'biomorphs' [30,31]. Carbonate nucleation onto the metal cations causes a local pH decrease, stimulating silica precipitation which raises the pH again and promotes further carbonate precipitation, and so on through an iterative feedback process of pH oscillation and mineral precipitation [31]. The resulting structures show diverse morphological complexity and provide favourable substrates for the condensation of organic carbon, leading to the production of convincing, partly organic pseudofossils comparable to controversial microstructures found in Archaean cherts [31]. Silica-carbonate biomorphs do not typically form long, untwisted filaments with circular cross-sections, hollow tubes with consistent diameters or lifelike, non-crystallographic bifurcation. More importantly, silica-carbonate biomorphs are not composed of iron minerals or metal (oxyhydr)oxides. These differences, together with a perception that mineral self-organization processes in general require 'exotic' or unusual chemistry, have been interpreted by some authors to militate against the origination of iron-mineral filaments in the rock record through such processes (e.g. [1,5,7]).

Here, I report the facile experimental production of microscopic filamentous iron-mineral biomorphs that replicate key morphological features previously thought to imply a biotic origin for compositionally similar filaments in the rock record. These structures result from the well-known phenomenon of chemical gardening (also known as 'silica gardening'), whereby the dissolution of a 'seed' metal salt into an alkaline carbonate or silicate solution produces a pocket of acidic fluid enclosed by a gelatinous membrane of hydrous metal carbonate or silicate together with metal (oxyhydr)oxides [32–35].

As osmotic inflow creates internal pressure, the membrane-bound pocket grows larger and eventually ruptures, at which point a jet of fluid is ejected from the breach and rapidly enclosed by a new, tube-like membrane, which continues to extend [32]. Further ruptures may occur in the walls or tips of these tubes, generating further, ramifying extensions until the internal–external pH gradient abates, yielding tubular filaments with lifelike morphologies. As hydroxyl ions flow into these structures, they react with the metal cations of the internal solutions to precipitate a coating of metal (oxyhydr)oxides on the inner walls of the tubes [32,33]. Tubes can diverge and (re)converge during growth, producing branches and anastomoses at any angle. Tubes may remain attached to a knob-like remnant of the membrane that formed around the original solid or detach from it spontaneously during growth. These processes have been studied for nearly a century and are increasingly well understood, yet their potential to form large populations of microscopic iron-mineralized filaments closely resembling purported fossil assemblages has rarely been recognized [28,36].

## 2. Results

### (a) Morphology and appearance

In this study, filamentous biomorphs (figure 1) were produced in less than 48 h from polycrystalline ferrous sulfate granules placed into aqueous solutions of sodium silicate or sodium carbonate at standard temperature and pressure. Elaborate, flexible transparent tubes appeared within 1–5 min, darkened from pale green to red-brown over 24–48 h, and became more brittle as $Fe^{2+}$ phases formed internally and then oxidized to $Fe^{3+}$ [34]. Tube diameter was controlled primarily by seed grain size; grains sieved under atmospheric conditions to less than 63 µm produced tubes of broadly consistent external diameter (figure 1a); diameters were unimodally distributed around a median of 3.9 µm, with a pronounced positive skew (min = 1.8 µm; max = 16.9 µm; $n$ = 200; electronic supplementary material, table S1 and figure S1). Individual tubes could, however, widen by an order of magnitude if they reached and flared out along the water–air interface. Silicate solutions produced filaments oriented in all directions but with an initial bias towards the vertical, especially if larger grains were used, whereas growth in carbonate solution was both vertical and horizontal (the latter along the base of the vessel). Filaments grown in silicate solution were resilient enough to withstand the removal or evaporative precipitation of the solution, remaining intact standing freely in the air or embedded in solid silicate, respectively; the former would be impossible to achieve had growth required a solid or highly viscous medium [4].

These biomorphs showed several features previously suggested to support biological explanations or even to eliminate non-biological explanations for the occurrence of morphologically similar iron-mineral filaments occurring in rocks, including both straight and strongly curved trajectories ([8,10]; figure 1a), both filled and unfilled interiors ([1,11,14]; figure 1b–d), circular cross-sections ([6,13]; figure 1b–d), multiple attachment to an individual knob ([1]; figure 1e), frequent true (non-crystallographic) branching (bifurcation) at high angles ([9,10,28]; figure 1e–g), rare anastomosis ([2,9]; figure 1h), rare nestedness ([1]; figure 1i), and discrete, spore-like swellings ([6,13,15]; figure 1j). Other notable features were a tendency to taper slightly at the ends of the filaments (figure 1j), frequent changes of filament growth direction (figure 1e), and occasional parallel growth of mutually adhesive filaments without convergence. Filaments and tubes grown in sodium carbonate tended to be less complex than those grown in sodium silicate, with stronger curvature

**3**

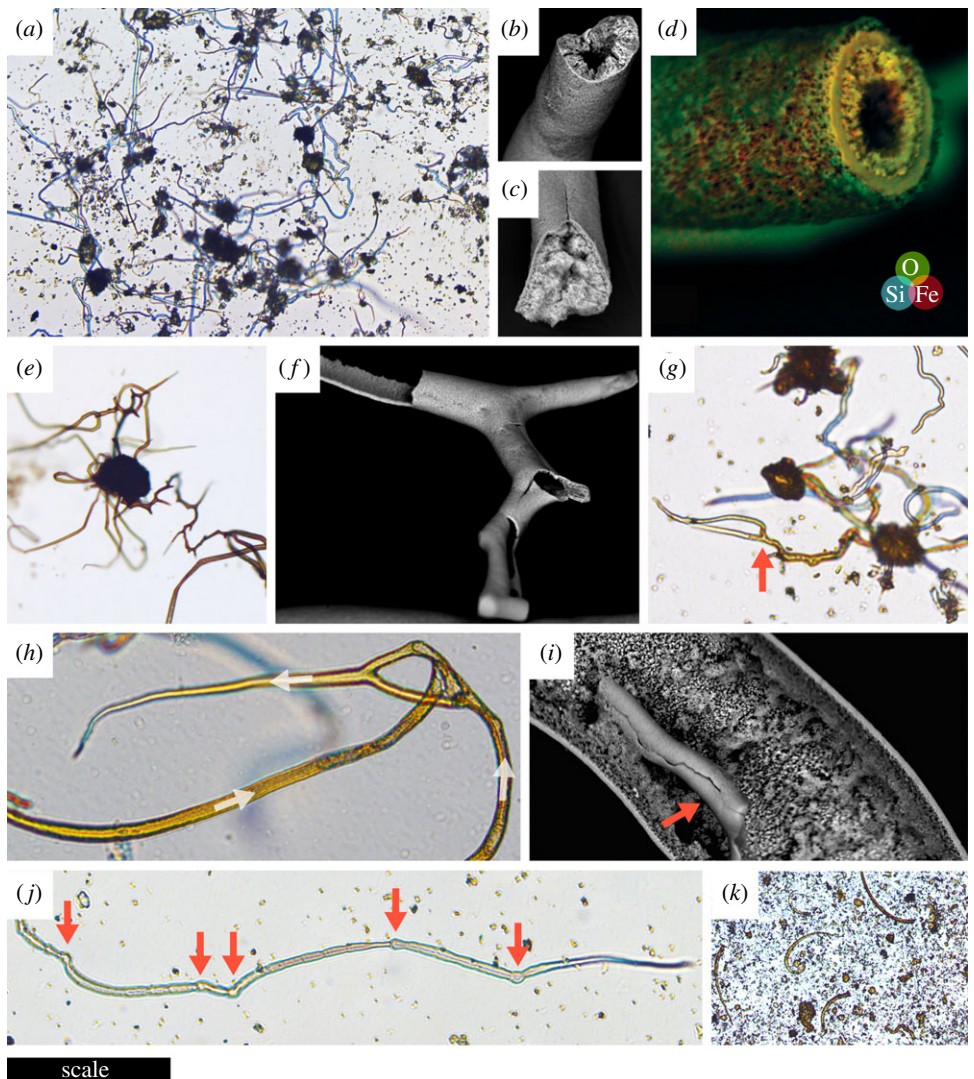

**Figure 1.** Photomicrographs and scanning electron micrographs of experimental iron-mineralizing chemical gardens. (*a*) Numerous straight and irregularly curved siliceous filaments attached to the knob-like remnants of iron sulfate seed grains less than 63 μm in diameter. (*b*) Siliceous filament showing rough iron (oxyhydr)-oxide-coated interior with hollow central cavity. (*c*) Siliceous filament with the central cavity filled by iron (oxyhydr)oxides. (*d*) Siliceous filament with laminated wall; overlaid energy-dispersive X-ray (EDX) spectroscopy data show iron-rich innermost layers (red/yellow). (*e*) Multiple siliceous branching filaments radiating from a seed grain remnant; the brown colour is contributed by ferric iron. (*f*) Branching siliceous tube with minimal inner coating. (*g*) Siliceous filaments with variable yellow/brown inner coating showing branch (arrowed). (*h*) Anastomosing siliceous filaments; arrows indicate direction of growth. (*i*) Broken siliceous filament (arrowed) on the interior of a larger tube. (*j*) Siliceous filament with discrete swellings (arrowed). (*k*) Curving filaments produced from iron sulfate seed grains in sodium carbonate solution. Scale bar: (*a*) 200 μm; (*b*) 95 μm; (*c*) 85 μm; (*d*) 45 μm; (*e*) 300 μm; (*f*) 115 μm; (*g*) 55 μm; (*h*) 55 μm; (*i*) 73 μm; (*j*) 70 μm; (*k*) 350 μm. (Online version in colour.)

and a greater tendency to tapering, but no unambiguous branching and rougher surface texture (figures 1*k* and 3).

## (b) Composition

The chemistry of chemical gardening elaborated by previous experimental studies [32–35] suggests that the filaments grown in the present experiments should be dominated by iron (oxyhydr)oxides, with amorphous hydrated Fe–Si phases also present initially in the exterior of filaments grown in silicate, although these can redissolve and are not necessarily integral to the final solid product [34]. These expectations were borne out by scanning electron microscopy (SEM)-EDX, Raman, and X-ray diffraction (XRD) analysis in the present study. EDX revealed that the rough material forming the filaments grown in carbonate and the interiors of those grown in silicate was iron and oxygen-rich, while the silicate-grown filaments also displayed smooth exteriors enriched in silicon (figure 1*b*,*c*). Raman spectra of filaments, whether grown in

sodium silicate or sodium carbonate solutions, initially showed indistinct or weak bands, probably due to a combination of high photosensitivity and poor crystallinity [38,39]. However, re-analysis of previously analysed spots consistently yielded narrow peaks at about 220, 240 290, 405, and 605 cm$^{-1}$ (figure 2*a*), closely comparable with standard reference spectra for hematite [37]. This suggests that iron (oxyhydr)oxides present in the sample underwent Raman laser-induced photolytic transformation to hematite, a phenomenon previously noted to occur with ferrihydrite, magnetite, and maghemite [38,40]. The X-ray diffractograms obtained from powdered filaments were noisy, with low-intensity peaks that were difficult to distinguish visually from the background, which is characteristic of poorly crystalline iron-rich material. However, automated comparison with the reference database showed that the XRD peaks are consistent with the occurrence of hematite, goethite, ferrihydrite, and feroxyhyte as crystalline phases present in the filaments grown both in sodium carbonate and sodium silicate solutions (figure 2*b*). Both ferrihydrite and

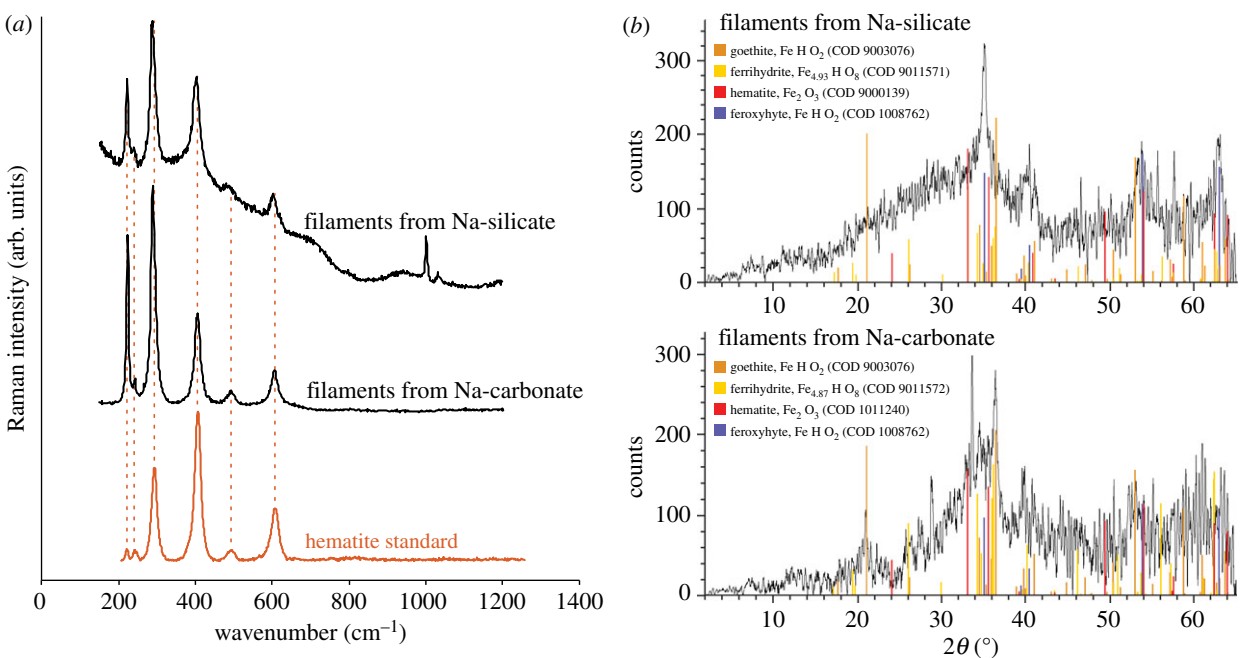

**Figure 2.** Composition of chemical garden filaments showing iron (oxyhydr)oxides. (*a*) Raman spectra showing characteristic peaks for hematite obtained on repeat analysis of Raman laser-damaged filaments. The additional peaks at approximately 1000 cm$^{-1}$ in the uppermost spectrum are due to the underlying plastic Petri dish. The hematite standard shown for comparison is RRUFF 040024 [37]. (*b*) XRD traces showing the occurrence of diffraction peaks at angles consistent with goethite, ferrihydrite, hematite, and feroxyhyte (corresponding reference sample numbers in the Crystallographic Open Database are indicated). The low signal-to-noise ratio is due to poor crystallinity and iron fluorescence. (Online version in colour.)

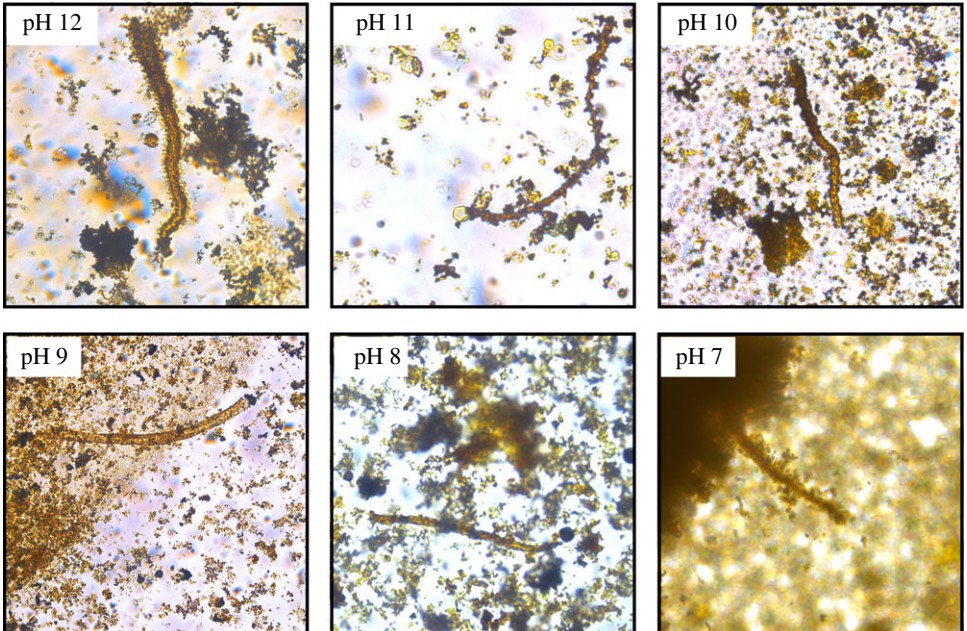

**Figure 3.** Photomicrographs of individual filaments grown in sodium carbonate solutions acidified to pH 12–7. The filaments illustrated are of lengths 200 µm (pH 12), 120 µm (pH 11), 120 µm (pH 10), 220 µm (pH 9), 105 µm (pH 8), and 120 µm (pH 7). (Online version in colour.)

feroxyhyte have previously been reported as products of the reaction between ferrous sulfate and sodium silicate solutions, in agreement with this interpretation of the XRD data [41].

## (c) Growth at lower solution pH

I observed the effect of lowered pH on the growth of iron-mineral filaments. In sodium carbonate solutions acidified with hydrochloric acid (thus containing less carbonate and more bicarbonate in solution), filaments and tubes grew abundantly from ferrous sulfate grains at pH values as low

as 9.0, and sparingly at pH values as low as 7.0, despite the paucity of carbonate ions at this pH (figure 3). Sodium silicate solutions polymerized too rapidly for experiments below pH 11.0, but at this pH filaments also grew abundantly, albeit more slowly than at higher pH.

## 3. Discussion

### (a) Comparison with previously reported biomorphs

Here, I have shown that the reaction of ferrous sulfate grains with sodium carbonate and sodium silicate solutions in shallow

vessels (Petri dishes, limiting vertical extension and introducing an effect of surface tension) allows for the rapid production of large populations of straight and curved filaments with consistently microbe-like sizes and morphologies, including circular cross-sections, non-crystallographic bifurcation during growth, anastomosis, and nestedness. Compositionally, these biomorphs are typical of iron-based chemical gardens previously described in the experimental literature. Most previous experimental studies following the 'classic' procedure have used salt granules or pellets several millimetres in diameter immersed beneath several centimetres of solution within test tubes or similar reaction vessels (e.g. [32–35]). This method produces vertically oriented, chimney-like structures several centimetres in length controlled by buoyancy-driven extension, commonly with several sub-vertical branches, which do not closely resemble candidate fossils in their overall morphology. Other studies have used vertically confined spaces to produce smaller, quasi-2D chemical gardens that form meandering filaments with infrequent branching [42,43]. Interestingly, one of these studies [43] describes self-avoidance during filament growth that would seem to preclude anastomosis. In the present study, anastomosis was present but rare, and filaments sometimes met and grew along with each other as if mutually adhesive but unable to converge into a single filament.

The present results do not exhaust the morphospace accessible to chemical gardens, which can also produce pseudoseptate filaments and spherical bulbous terminations resembling fungal sporangia (e.g. figures 38, 39, 48, 55, and 56 in [44]). Serially twisted/helical filaments or 'stalks' of iron oxide, which are widely regarded as biosignatures for iron-oxidizing bacteria (e.g. [28]) were not produced in the present study, but classic work suggests that serially twisted forms can also occur (e.g. figure 55a in [44]). Silica-carbonate biomorphs also show helical forms, and further extend the morphospace of abiotic mineral growth structures to encompass fractally branching dendrites, framboid-like masses, rope-like twisted threads and ribbons, and complex shapes resembling urns, corals, and snails, but not closely resembling the iron-mineral filaments addressed by this study [45,46].

## (b) Comparison with iron-mineral filaments in the rock record

Iron-mineral filament assemblages previously interpreted as fossilized microbial populations are composed largely of hematite (e.g. [1,2,7,8,11,15]), iron oxyhydroxides such as goethite and ferrihydrite (e.g. [2,5,7,8,10–12,28]), and iron-rich aluminosilicate clay minerals [6–8,10,25]. In line with previous studies of tubular chemical gardens using iron salts [30,32,33], the Raman, EDX, and XRD analyses in the present study suggest that biomorphs produced by reacting ferrous sulfate with either sodium carbonate or sodium silicate solutions were composed largely of iron oxyhydroxides. These minerals very readily transform to hematite during diagenesis or metamorphism, and may also serve as precursors to Fe-rich phyllosilicates in hydrothermal, silica-rich settings [47,48]. Bacterial iron oxidation can likewise produce iron (oxyhydr)oxides (e.g. [49]), but the present results show that the composition of iron-mineral filaments in the rock record is equally consistent with origination through abiotic processes. A recent study also interpreted hollow silica tubes from hydrothermal deposits of the Arctic Mid-Ocean Ridge as possible chemical gardens [28]; such tubes may be obtainable from experiments like those reported here if terminated before iron oxyhydroxides encrust the initial siliceous membranes; cf. the outer layer in figure 1d [32,33].

When produced from seed grains sieved to less than 63 μm in diameter, 188 out of 200 individual chemical garden filaments measured in this study showed external diameters between 2 and 10 μm (median 3.9 μm); no filaments were narrower than 1 μm, and only four were wider than 12 μm (electronic supplementary material, figure S1). This size distribution and range are similar to numerous assemblages of iron-mineral filaments in the rock record (e.g. [1,6,7,50–52]). The chemical gardens in this study (figure 1) also reproduce almost the full range of morphological characteristics (straight and curved trajectories with changes of direction; filled and unfilled (hollow) interiors; circular cross-sections; multiple attachment to knobs; discrete swellings; non-crystallographic, constant-thickness branching; anastomosis; nestedness) previously thought to show that naturally occurring iron-mineral filaments are likely to be microfossils (e.g. [1,2,4,6–15]). It is important to concede that I did not produce true septate filaments with internal, walled compartments, a feature which has been observed in some natural iron-mineral filament assemblages where carbonaceous residues provide additional evidence for biogenicity (e.g. [9,19,20]). Additionally, although filament thickness was usually conserved during growth even during branching and anastomosis (e.g. figure 1e,f,g), this was not always the case; bifurcation could reduce filament thickness while re-convergence could increase it, leading to some dubiously lifelike morphologies, especially in larger filaments. In addition, some filaments tapered gradually in the direction of growth (e.g. figure 1j).

These results are strikingly similar to the assemblage of hematite tubes and non-septate filaments in hydrothermal chert beds of the $4.0 \pm 0.3$ giga-annum (Ga) Nuvvuagittuq Greenstone Belt, northeast Canada, recently interpreted as Earth's oldest body fossils [1]. The filaments are reportedly 2–14 μm in diameter and up to 500 μm in length, and have been interpreted as the partly permineralized, partly encrusted remains of iron-oxidizing bacteria [1]. Some filaments are attached to knobs 80–120 μm in diameter, and some are nested within tubes (16–30 μm in diameter and 80–400 μm in length), which also occur without filaments; these features were considered incompatible with an abiotic origin, but are replicated abiotically in the present study (figure 1e,i). Buoyancy- or flow-driven growth of chemical gardens from fairly uniform parent crystals or grains would also explain the straight, unbranched, parallel nature of some of the Nuvvuagittuq tubes and their consistent sizes. Hollow tubes could also have originated via dissolution, diffusion, and re-precipitation of filaments during the polymerization of the surrounding silica, with or without leaving residual filaments inside; filaments in some moss agates are surrounded by (commonly multiple) concentric sheath-like tubes likely to have formed similarly [53,54]. Other evidence adduced to support the biogenicity of the Nuvvuagittuq filaments (e.g. the presence near the filaments of graphite, carbonate rosettes with isotopically light carbon, and phosphate) does not settle the biogenicity of the filaments themselves, which are morphologically simple and strictly non-carbonaceous. It is not implausible that alkaline

fluids generated by serpentinization of the mafic (sub)sea-floor promoted the growth of chemical gardens in this setting.

The results are also reminiscent of numerous candidate microfossils proposed to have formed in subsurface environments, i.e. the deep biosphere (e.g. [5–10,50,51]; see review in [18]). Among these, one assemblage of special scientific importance is the suite of iron-rich chloritic filaments preserved within calcite- and chlorite-filled amygdales (mineralized vesicles) in basalts from the lower part of the 2.4 Ga Ongeluk Formation of South Africa [6]. These filaments were recently interpreted as the oldest fossil eukaryotes, but are similar to the chemical gardens described in the present study in several respects. They are solid, apparently non-septate, about 2–12 µm in diameter, and up to hundreds of µm in length. They are composed of iron-rich chlorite, a common vein- and amygdale-filling phyllosilicate in hydrothermally altered basaltic rocks, where it also forms the filamentous dubiofossil 'moss' found in moss agates [55]. The origin of the Ongeluk chlorite is not precisely known; it could derive from the alteration of smectite that replaced organic matter as proposed by Bengtson *et al.* [6], but smectite can also form via the interaction of hydrothermal silica and iron oxyhydroxides, i.e. the constituents of chemical garden filaments [48]. Independent evidence for an influx of silica-rich hydrothermal fluids exists in the lower part of the Ongeluk Formation in the form of abundant hydrothermal jasper and chert deposits [56].

While the composition of the Ongeluk filaments is seemingly compatible with both biotic and abiotic interpretations, the argument that they are biotic rests largely on their morphological and organizational resemblance to putative fossil fungi from much younger rocks (including some that preserve organic matter). The Ongeluk filaments show curvilinear trajectories, branching, anastomosis, circular cross-sections, and bulbous protrusions. The results of the present study show that all these features are equally consistent with chemical garden growth. Neither the radiating growth of filaments inwards from cavity walls (also seen in moss agates) nor the occurrence of multifurcate, entangled 'broom' structures [6] was replicated in my Petri dish experiments, but these features do not seem fundamentally incompatible with chemical garden growth provided with the appropriate distribution of seed material and the correct flow regime and rate. Chemical garden filaments are flexible in the early, gelatinous phase of growth and can become entangled during growth with or without anastomosing. The irregular chlorite lining Ongeluk amygdales, described by Bengtson *et al.* [6] as a 'basal film consisting of a jumbled mass', could represent an amalgamation of the membranes formed around seed material in chemical gardens, which become mineralized along with the filaments (figure 1*e*,*g*). More naturalistic experimental systems must be used to test these proposals before the hypothesis that the Ongeluk filaments represent chemical gardens can be evaluated fully.

## (c) Plausibility of chemical garden growth in nature

Chemical gardens are already thought to occur in geological settings where silica and/or carbonate-laden alkaline fluids react with metalliferous mineral particles or solutions, most notably forming complex structures at marine hydrothermal vents (e.g. [28]; see also [32] for a discussion of chemical gardens in nature). Deep, isolated groundwater tends to become somewhat alkaline (as well as carbonate- and silica-rich) as a consequence of water–rock reactions that consume $H^+$, and in some settings the hydrolysis of olivine and pyroxene in basalts and ultramafic rocks (serpentinization) leads to groundwater pH values as high as 10–12.6 [48,57–61]. Lakes fed by hydrothermal systems in the East African Rift Valley are sufficiently alkaline and silica-rich to be theoretically compatible with biomorph production at the Earth's surface [46], and it has recently been demonstrated experimentally that naturally occurring silica-rich alkaline spring waters are capable of inducing the growth of classical chemical gardens from iron salts, as well as producing silica-carbonate biomorphs [35]. Moreover, the results presented here show that very high pH is not required to form microbe-like filaments, which grew in sodium carbonate solutions acidified to mildly alkaline and even neutral pH (figure 3). Thus, it is reasonable to suppose that groundwater in many of the settings where iron-mineral filament assemblages have been found—silicifying/calcifying marine hydrothermal systems, volcanic rocks near mid-ocean ridges and deeply buried on land, and limestones—could have become sufficiently alkaline to precipitate iron-mineral chemical garden filaments. Further experimental work is, however, needed to test this supposition. Since naturally occurring iron-mineral filaments are widely associated with the common ferrous sulfide mineral, pyrite (e.g. [4,7,62]), I further speculate that the ferrous sulfate minerals or solutions derived from the oxidation of iron sulfide minerals (not necessarily abiotically) may have stimulated the formation of filamentous chemical gardens in some natural settings (a pyrite precursor for some moss agates was also suggested by Hopkinson *et al.* [27]).

## (d) Discriminating between iron-mineralized chemical gardens and fossil microbes

Some natural iron-mineral filament assemblages contain complex organic matter and phosphate, together with iron-mineral growth-textures strongly suggestive of encrustation onto pre-existing organic material, implying that they are more likely to be fossils than not (e.g. [63,64]). Filaments associated with carbonaceous material of indeterminate origin are not necessarily biogenic [31], and most iron-mineral filament assemblages lack such material altogether. Nevertheless, iron-encrusted microbial filaments and abiotic chemical garden filaments and tubes are unlikely to be perfectly indistinguishable in composition, morphology, texture, or organization at all scales, and the possibility remains that diagnostic differences may be discovered [28]. Statistical analyses of morphometric parameters over large populations of biotic and abiotic filaments may be fruitful; preliminary steps have been taken in this direction (e.g. [8,28,45,52]). The controlled experimental iron-mineral encrustation of large numbers of bacterial and fungal filaments will be necessary to provide suitable datasets. As a corollary, experiments to grow chemical gardens in the presence of filamentous microbes may be worthwhile in case this leads to new morphologies. Submicroscopic internal and external textures of biotic and abiotic filaments, not explored in detail by the present work, should be compared. Both smooth-walled and more coarsely crystalline tubes and filaments are found in natural iron-mineral filament assemblages, even together within the same assemblage (e.g. [1]). In the present study, abiotic filaments grown in sodium

silicate solution showed smoother exteriors than those produced in sodium carbonate. Smoothness has recently been shown to respond to growth rate, with slow-forming chemical garden filaments tending to show more coarsely textured walls [65]; it has also been shown that chemical gardens grown from ferrous chloride differ microtexturally (and mineralogically) from their ferric equivalents [34]. It was recently pointed out [28] that concurrent precipitation of silica and iron minerals might produce a diagnostically abiotic internal structure in some natural filament assemblages, i.e. a diffuse filament core zone composed of iron-mineral spherules supported by a silica matrix; this was not observed in the present study, but might perhaps occur if more highly polymerized silica media were used.

## 4. Conclusion

This work has shown that the self-organizing behaviour of pH-driven inorganic chemical reactions can produce iron-mineralized filaments and tubes closely analogous in both morphology and composition to numerous microstructures found in diverse rocks of all ages, which have hitherto been interpreted as fossil microorganisms. It would be rash to conclude that these geologic assemblages exclusively represent mineralized chemical gardens, but the present results show that this abiotic mode of origin must be considered as a plausible 'null hypothesis' to be rejected or not on a case-by-case basis depending on the evidence. Future work may yet discover generalizable differences between biotic iron-encrusted filaments and abiotic iron-mineral microstructures. Such differences should be sought in the morphology, internal texture, wall texture, and spatial arrangements (distributions and orientations) of filaments. Meanwhile, the evidence presented here lends weight to previous exhortations not to regard morphological evidence alone as conclusive in determining the biogenicity of lifelike microstructures (e.g. [21]). I conclude that on the present state of knowledge, iron-mineral filaments selected on morphological grounds would make a questionable sample-return target for evidence of a biosphere on early Mars, where chemical gardens may well have formed in the presence of alkaline, silica-rich groundwater [46,66].

## 5. Materials and methods

### (a) Growth of chemical gardens

Unless otherwise indicated, the chemical gardens reported here were produced by manually dispersing polycrystalline granules (seed grains) of iron (II) sulfate heptahydrate (98% $FeSO_4 \cdot 7H_2O$; Alfa Aesar, Heysham, UK) into 15 ml solutions of sodium silicate or sodium carbonate (or mixtures thereof) in lidded (unsealed) Petri dishes at room temperature and pressure.

Alkaline solutions were prepared using distilled water (Thermo Scientific Barnstead Nanopure) and $100 \text{ g l}^{-1}$ of either:

(1) sodium silicate powder (53% $SiO_2$, 26% $Na_2O$, Scientific Laboratory Supplies, Nottingham, UK); measured pH was 12.4 except for one experiment, in which pH was adjusted with $HCl_{(aq)}$ to approximately 11.1 (at lower pH, the silica polymerized too rapidly); or

(2) sodium carbonate powder (99.6% $Na_2CO_3$, Acros Organics, Geel, Belgium); measured pH was 12.0 except for five additional experiments (figure 3), in which pH was adjusted with $HCl_{(aq)}$ to pH values of 7–11 (thus also removing carbonate ions and forming a quantity of NaCl).

A Jenway 3510 pH Meter calibrated with standard solutions at pH 4.0, 7.0, and 10.0 was used to measure pH. Filaments were rinsed four times in distilled water and dried in air before compositional analyses were undertaken.

### (b) Scanning electron and optical microscopy and energy dispersive X-ray spectroscopy

The backscattered electron micrographs and EDX data shown in figure 1b–d,f, and i were obtained with a Carl Zeiss SIGMA HD VP Field Emission scanning electron microscope fitted with an Oxford AZtec ED X-ray analysis system. The remaining panels in figures 1 and 3 were obtained with a Leica DM RP microscope.

### (c) X-ray diffraction

Experimental products were ground by hand in an agate mortar to homogenize them and reduce their grain size to approximately 50 μm or less. Samples were then loaded into polycarbonate sample holders. Care was taken to achieve a planar sample surface with a minimum of compression to lessen the effects of preferred orientation in the sample pile. The samples were scanned in a Bruker D8-Advance X-ray Diffractometer in the School of Geosciences, University of Edinburgh, employing a Bragg-Brentano source-sample-detector configuration. The primary X-rays were generated by a Cu-anode X-ray tube operating at an excitation voltage of 40 kV and a tube current of 40 mA. The diffracted X-rays were detected using a NaI scintillation detector filtered using a 700 μm thick Ni filter to eliminate lines generated by the Cu Kb radiation. The samples were scanned from 2° to 70° 2θ at a step size of 0.025° and dwell time of 6 s per step. The traces were analysed using the Bruker EVA database and peaks assigned using the internal crystallographic open database (COD) database.

### (d) Raman spectrometry

Raman spectra were acquired with an inVia Raman system (Renishaw plc) coupled to a Leica DMLM microscope at the University of Edinburgh. The 785 nm (300 mW) excitation laser beam (Toptica) was focused onto the samples using a ×100/0.9 NA objective lens (Leica, HCX PL Fluotar), providing an excitation spot of 1 μm diameter. Raman point spectra were taken at different positions on the samples over the range $100–1200 \text{ cm}^{-1}$ in extended scan mode. The spectra were acquired with 30 s exposure time using a $600 \text{ lines /mm}^{-1}$ diffraction grating. Wire 2.0 software was used for data acquisition.

Data accessibility. All data are available in the main text or electronic supplementary material.

Competing interests. The author declares no competing interests.

Funding. This work was funded by the European Union's Horizon 2020 Research and Innovation Programme under Marie Skłodowska-Curie grant agreement no. 747877.

Acknowledgements. I thank C. S. Cockell for helpful discussions, A. McDonald for assistance with Raman spectroscopy, N. Cayzer for assistance with SEM, and N. Odling for assistance with XRD. This manuscript was improved by the comments of Nicola McLoughlin and two anonymous reviewers.

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
