## [Reviewer comments · Proceedings of the Royal Society B: Biological Sciences]

Review History

RSPB-2019-1781.R0 (Original submission)

Review form: Reviewer 1

Recommendation

Major revision is needed (please make suggestions in comments)

Scientific importance: Is the manuscript an original and important contribution to its field?

Acceptable

General interest: Is the paper of sufficient general interest?

Good

Quality of the paper: Is the overall quality of the paper suitable?

Good

Is the length of the paper justified?

Yes

Should the paper be seen by a specialist statistical reviewer?

No

Do you have any concerns about statistical analyses in this paper? If so, please specify them explicitly in your report.

No

It is a condition of publication that authors make their supporting data, code and materials available - either as supplementary material or hosted in an external repository. Please rate, if applicable, the supporting data on the following criteria.

Is it accessible?

Yes

Is it clear?

Yes

Is it adequate?

No

Do you have any ethical concerns with this paper?

No

Comments to the Author

This manuscript addresses a crucial question in geobiology: the difficulties of distinguishing between non-biogenic and biogenic objects of simple morphologies. Life-like morphologies can be produced by physicochemical processes, and one well-known example is the “chemical gardens” often sold as toys, which produce intricate plant-like patterns by the addition of metal salts to a silicate solution. These patterns are the result of complex interplays between osmosis and precipitation. The relevance for the interpretation of putative microbial fossils has been studied in particular by the research group of J.M. García-Ruiz, who has demonstrated various life-like shapes, including “leaves” and “cauliflowers”, as well as filaments of different dimensions that may be straight, curved, simple, branching, but usually helicoidal.

Although these phenomena are known and typically cited in studies of ancient putative microbial fossils, interpretations of the fossils are frequently controversial on account of the biogenicity problem. Also even where there is geochemical evidence from organic carbon in support, this is often not enough, since evidence of biological activity does not necessarily mean that the studied morphologies are biologically generated.

McMahon presents experiments producing ferric-encrusted filaments by seeding ferrous sulfate granules (<63 μm) in silicate or carbonate solutions, a version of chemical gardens. The growth space was vertically restricted by the use of Petri dishes. The obtained structures resemble filamentous tubular biomorphs previously reported from chemical gardens by, e.g., Haudin et al. 2015 (PCCP 17:12804–12811) and Brau et al. 2018 (PCCP 20:784–793). These authors used a vertically confined space of 0.5 mm for their experiments. McMahon’s structures are more life-like in that they are more frequently branching; otherwise there is little to distinguish them morphologically. Similarly, the experimentally produced minute tubes reported by Makki et al. 2009 (Angewandte Chemie International Edition 48:8752–8756) are similar in dimensions and morphology, although they do not show branching. In particular, Makki et al. show how fast growth produces smooth and comparatively straight tubes, whereas slower growth results in tubes that appear grainy and segmented. None of the references above are in McMahon’s manuscript.

McMahon's claim for anastomoses, a salient feature of fungal mycelia and of the Palaeoproterozoic Ongeluk fungus-like structures discussed in the manuscript, seems poorly founded. The alleged anastomoses are said to occur "at any angle" but to be rare, and the only figured example (Fig. 1H) is singularly unconvincing, as it shows only a minor detachment from and reunification with the same filament. This might not even be a growth feature, but an instance of parallel fibres being somewhat teased apart. A true anastomosis implies a branch seeking out a filament other than that which it originates from, and merging with it, and this is not shown in Fig. 1H. Significantly, the filaments reported by Haudin et al. and Brau et al. (refs above) showed a clear self-avoidance, seemingly precluding the formation of anastomoses.

There are other features of the Ongeluk structures that remain unexplained by the chemical gardens. In the Ongeluk vesicles, the filaments form a jumbled mat adhering to the rock walls which then sends out filaments into the voids. This behaviour is neither observed nor readily explained in a chemical model. The Ongeluk filaments are frequently entangled in a way that indicates original flexibility, and they tend to form broom-like aggregates. This habitus has still to be demonstrated in chemical-garden experiments.

The Archean/Hadean Nuvvuagittuq structures, like the Ongeluk ones, lack chemical evidence of an originally organic composition, and they are morphologically simpler than the latter. McMahon's discussion of biogenicity in the Nuvvuagittuq is more to the point, and the arguments temper the cautiously positive reception these structures have received in the literature, e.g., Whitehouse et al. 2019 (Precambrian Research 323:70–81).

The experimental results in the manuscript add to the known variety of biomorphs produced by abiotic processes. Eventually the paper should be published, but it must then include a fuller treatment of the literature (cf. the references given above that present results similar to the ones in the manuscript), and the important claim about anastomosing filaments must be backed up by documentation or dropped altogether. Arguments for an abiogenic origin of the Ongeluk filaments need to be made stronger or dropped.

A stylistic point: A single author should not refer to himself as "we".

Review form: Reviewer 2 (Nicola McLoughlin)

Recommendation

Major revision is needed (please make suggestions in comments)

Scientific importance: Is the manuscript an original and important contribution to its field?

Excellent

General interest: Is the paper of sufficient general interest?

Good

Quality of the paper: Is the overall quality of the paper suitable?

Good

Is the length of the paper justified?

Yes

Should the paper be seen by a specialist statistical reviewer?

No

Do you have any concerns about statistical analyses in this paper? If so, please specify them explicitly in your report.

No

It is a condition of publication that authors make their supporting data, code and materials available - either as supplementary material or hosted in an external repository. Please rate, if applicable, the supporting data on the following criteria.

Is it accessible?

Yes

Is it clear?

Yes

Is it adequate?

No

Do you have any ethical concerns with this paper?

No

Comments to the Author

Please e-mail me for the Johannessen et al (in press) article as it is too large to upload to the reviewer website. (See Appendix A)

Review form: Reviewer 3

Recommendation

Accept with minor revision (please list in comments)

Scientific importance: Is the manuscript an original and important contribution to its field?

Excellent

General interest: Is the paper of sufficient general interest?

Good

Quality of the paper: Is the overall quality of the paper suitable?

Excellent

Is the length of the paper justified?

Yes

Should the paper be seen by a specialist statistical reviewer?

No

Do you have any concerns about statistical analyses in this paper? If so, please specify them explicitly in your report.

No

It is a condition of publication that authors make their supporting data, code and materials available - either as supplementary material or hosted in an external repository. Please rate, if applicable, the supporting data on the following criteria.

Is it accessible?

Yes

Is it clear?

Yes

Is it adequate?

Yes

Do you have any ethical concerns with this paper?

No

Comments to the Author

This work is very important, since abiotic processes may form chemical and morphological features similar to biological processes. This fact may result in misinterpretation of fossils and biosignatures if care is not taken. So, this work is therefore an important contribution to the research fields on early life on earth and fossilized microorganisms. The paper is well structured and concise. I recommend accept with a few minor revisions.

Comments:

* It would be interesting to see some more in depth comparative discussion on the papers by Glaab et al. 2016 or Opel et al. 2019, where they use iron salts and lower pH. The paper by Glaab is mentioned but a comparison with this work would be very interesting, I think.

* Are there any natural environments today where biomorphs have been found? The authors talk about plausibility of chemical garden growth in nature but do not give actual examples of places where any of these gardens are found? Except for fossilized structures? Are there any places where chemical gardens are forming these structures today? It would be helpful with some concrete examples on where to look. There are examples such as a) deep, isolated groundwaters and b) silica-rich alkaline springs but no evidence on actual chemical gardens have been found there yet. If not, why? Are there other places to look for those structures? What would such sampling require? The authors discuss briefly on these matters but it would be interesting to know more and a bit more in depth.

* What kind of further experiments would be needed to distinguish between fossilized abiotic and biologic iron oxide tubes? Tagged samples? Are there examples of similar experiments that of the authors of this manuscript, but with added microorganisms? If so, are there differences in the morphology between abiotically and biologically formed iron oxide tubes?

Decision letter (RSPB-2019-1781.R0)

29-Aug-2019

Dear Dr McMahan:

I am writing to inform you that your manuscript RSPB-2019-1781 entitled "Earth's earliest and deepest purported fossils may be iron-mineralized chemical gardens" has, in its current form, been rejected for publication in Proceedings B.

This action has been taken on the advice of referees, who have recommended that substantial revisions are necessary. With this in mind we would be happy to consider a resubmission, provided the comments of the referees are fully addressed. However please note that this is not a provisional acceptance. Indeed, all three reviewers (two in particular) have some issues with the science and are not certain the conclusions are as robust or novel as they are claimed; they will need more convincing if the MS is to be accepted in re-review.

In your revision process, please take a second look at how open your science is; our policy is that all data involved with the study should be made openly accessible-- see: <https://royalsociety.org/journals/ethics-policies/data-sharing-mining/>
Insufficient sharing of data can delay or even cause rejection of a paper.

Sincerely,

Professor John Hutchinson, Editor
mailto: proceedingsb@royalsociety.org

Associate Editor
Board Member: 1
Comments to Author:
Dear Sean

As you will see. The reviewers have made a number of comments and suggestions. There is some specific claims that needs to be further validated and considered. Also, there is a manuscript in press which you need to also consider in your writing.

Please have a look at these and decide if your manuscript still has the same validity in this new light and resubmit your manuscript, which will be sent back to the same referees.

Reviewer(s)' Comments to Author:

Referee: 1

Comments to the Author(s)

This manuscript addresses a crucial question in geobiology: the difficulties of distinguishing between non-biogenic and biogenic objects of simple morphologies. Life-like morphologies can be produced by physicochemical processes, and one well-known example is the “chemical gardens” often sold as toys, which produce intricate plant-like patterns by the addition of metal salts to a silicate solution. These patterns are the result of complex interplays between osmosis and precipitation. The relevance for the interpretation of putative microbial fossils has been studied in particular by the research group of J.M. García-Ruiz, who has demonstrated various life-like shapes, including “leaves” and “cauliflowers”, as well as filaments of different dimensions that may be straight, curved, simple, branching, but usually helicoidal.

Although these phenomena are known and typically cited in studies of ancient putative microbial fossils, interpretations of the fossils are frequently controversial on account of the biogenicity problem. Also even where there is geochemical evidence from organic carbon in support, this is often not enough, since evidence of biological activity does not necessarily mean that the studied morphologies are biologically generated.

McMahon presents experiments producing ferric-encrusted filaments by seeding ferrous sulfate granules (<63 μm) in silicate or carbonate solutions, a version of chemical gardens. The growth space was vertically restricted by the use of Petri dishes. The obtained structures resemble filamentous tubular biomorphs previously reported from chemical gardens by, e.g., Haudin et al. 2015 (PCCP 17:12804–12811) and Brau et al. 2018 (PCCP 20:784–793). These authors used a vertically confined space of 0.5 mm for their experiments. McMahon’s structures are more life-like in that they are more frequently branching; otherwise there is little to distinguish them morphologically. Similarly, the experimentally produced minute tubes reported by Makki et al. 2009 (Angewandte Chemie International Edition 48:8752–8756) are similar in dimensions and morphology, although they do not show branching. In particular, Makki et al. show how fast growth produces smooth and comparatively straight tubes, whereas slower growth results in tubes that appear grainy and segmented. None of the references above are in McMahon’s manuscript.

McMahon’s claim for anastomoses, a salient feature of fungal mycelia and of the Palaeoproterozoic Ongeluk fungus-like structures discussed in the manuscript, seems poorly founded. The alleged anastomoses are said to occur “at any angle” but to be rare, and the only figured example (Fig. 1H) is singularly unconvincing, as it shows only a minor detachment from

and reunification with the same filament. This might not even be a growth feature, but an instance of parallel fibres being somewhat teased apart. A true anastomosis implies a branch seeking out a filament other than that which it originates from, and merging with it, and this is not shown in Fig. 1H. Significantly, the filaments reported by Haudin et al. and Brau et al. (refs above) showed a clear self-avoidance, seemingly precluding the formation of anastomoses.

There are other features of the Ongeluk structures that remain unexplained by the chemical gardens. In the Ongeluk vesicles, the filaments form a jumbled mat adhering to the rock walls which then sends out filaments into the voids. This behaviour is neither observed nor readily explained in a chemical model. The Ongeluk filaments are frequently entangled in a way that indicates original flexibility, and they tend to form broom-like aggregates. This habitus has still to be demonstrated in chemical-garden experiments.

The Archean/Hadean Nuvvuagittuq structures, like the Ongeluk ones, lack chemical evidence of an originally organic composition, and they are morphologically simpler than the latter. McMahon's discussion of biogenicity in the Nuvvuagittuq is more to the point, and the arguments temper the cautiously positive reception these structures have received in the literature, e.g., Whitehouse et al. 2019 (Precambrian Research 323:70–81).

The experimental results in the manuscript add to the known variety of biomorphs produced by abiotic processes. Eventually the paper should be published, but it must then include a fuller treatment of the literature (cf. the references given above that present results similar to the ones in the manuscript), and the important claim about anastomosing filaments must be backed up by documentation or dropped altogether. Arguments for an abiogenic origin of the Ongeluk filaments need to be made stronger or dropped.

A stylistic point: A single author should not refer to himself as "we".

Referee: 2

Comments to the Author(s)
See attached PDF.

Referee: 3

Comments to the Author(s)

This work is very important, since abiotic processes may form chemical and morphological features similar to biological processes. This fact may result in misinterpretation of fossils and biosignatures if care is not taken. So, this work is therefore an important contribution to the research fields on early life on earth and fossilized microorganisms. The paper is well structured and concise. I recommend accept with a few minor revisions.

Comments:

* It would be interesting to see some more in depth comparative discussion on the papers by Glaab et al. 2016 or Opel et al. 2019, where they use iron salts and lower pH. The paper by Glaab is mentioned but a comparison with this work would be very interesting, I think.

* Are there any natural environments today where biomorphs have been found? The authors talk about plausibility of chemical garden growth in nature but do not give actual examples of places where any of these gardens are found? Except for fossilized structures? Are there any places where chemical gardens are forming these structures today? It would be helpful with some

concrete examples on where to look. There are examples such as a) deep, isolated groundwaters and b) silica-rich alkaline springs but no evidence on actual chemical gardens have been found there yet. If not, why? Are there other places to look for those structures? What would such sampling require? The authors discuss briefly on these matters but it would be interesting to know more and a bit more in depth.

* What kind of further experiments would be needed to distinguish between fossilized abiotic and biologic iron oxide tubes? Tagged samples? Are there examples of similar experiments that of the authors of this manuscript, but with added microorganisms? If so, are there differences in the morphology between abiotically and biologically formed iron oxide tubes?

Author's Response to Decision Letter for (RSPB-2019-1781.R0)

See Appendix B.

RSPB-2019-2410.R0

Review form: Reviewer 3

Recommendation

Accept as is

Scientific importance: Is the manuscript an original and important contribution to its field?

Excellent

General interest: Is the paper of sufficient general interest?

Excellent

Quality of the paper: Is the overall quality of the paper suitable?

Excellent

Is the length of the paper justified?

Yes

Should the paper be seen by a specialist statistical reviewer?

No

Do you have any concerns about statistical analyses in this paper? If so, please specify them explicitly in your report.

No

It is a condition of publication that authors make their supporting data, code and materials available - either as supplementary material or hosted in an external repository. Please rate, if applicable, the supporting data on the following criteria.

Is it accessible?

N/A

Is it clear?

Yes

Is it adequate?

Yes

Do you have any ethical concerns with this paper?

No

Comments to the Author

I think the author have addressed all my comments. I therefore recommend the manuscript to be accepted as is.

Decision letter (RSPB-2019-2410.R0)

28-Oct-2019

Dear Dr McMahon

I am pleased to inform you that your Review manuscript RSPB-2019-2410 entitled "Earth's earliest and deepest purported fossils may be iron-mineralized chemical gardens" has been accepted for publication in Proceedings B. Congratulations!!

The referee(s) do not recommend any further changes. Therefore, please proof-read your manuscript carefully and upload your final files for publication. Because the schedule for publication is very tight, it is a condition of publication that you submit the revised version of your manuscript within 7 days. If you do not think you will be able to meet this date please let me know immediately.

To upload your manuscript, log into <http://mc.manuscriptcentral.com/prsb> and enter your Author Centre, where you will find your manuscript title listed under "Manuscripts with Decisions." Under "Actions," click on "Create a Revision." Your manuscript number has been appended to denote a revision.

You will be unable to make your revisions on the originally submitted version of the manuscript. Instead, upload a new version through your Author Centre.

- 1) A text file of the manuscript (doc, txt, rtf or tex), including the references, tables (including captions) and figure captions. Please remove any tracked changes from the text before submission. PDF files are not an accepted format for the "Main Document".
- 2) A separate electronic file of each figure (tiff, EPS or print-quality PDF preferred). The format should be produced directly from original creation package, or original software format. Please note that PowerPoint files are not accepted.
- 3) Electronic supplementary material: this should be contained in a separate file from the main

text and the file name should contain the author's name and journal name, e.g
 authorname_procb_ESM_figures.pdf

All supplementary materials accompanying an accepted article will be treated as in their final form. They will be published alongside the paper on the journal website and posted on the online figshare repository. Files on figshare will be made available approximately one week before the accompanying article so that the supplementary material can be attributed a unique DOI. Please see: <https://royalsociety.org/journals/authors/author-guidelines/>

4) Data-Sharing and data citation

It is a condition of publication that data supporting your paper are made available. Data should be made available either in the electronic supplementary material or through an appropriate repository. Details of how to access data should be included in your paper. Please see <https://royalsociety.org/journals/ethics-policies/data-sharing-mining/> for more details.

If you wish to submit your data to Dryad (<http://datadryad.org/>) and have not already done so you can submit your data via this link <http://datadryad.org/submit?journalID=RSPB&manu=RSPB-2019-2410> which will take you to your unique entry in the Dryad repository.

Once again, thank you for submitting your manuscript to Proceedings B and I look forward to receiving your final version. If you have any questions at all, please do not hesitate to get in touch.

Sincerely,
 Professor John Hutchinson, Editor
<mailto:proceedingsb@royalsociety.org>

Associate Editor
 Board Member
 Comments to Author:
 Reviewer is happy with revisions. It is good to go.

Reviewer(s)' Comments to Author:

Referee: 3

Comments to the Author(s).

I think the author have addressed all my comments. I therefore recommend the manuscript to be accepted as is.

Decision letter (RSPB-2019-2410.R1)

31-Oct-2019

Dear Dr McMahon

I am pleased to inform you that your manuscript entitled "Earth's earliest and deepest purported fossils may be iron-mineralized chemical gardens" has been accepted for publication in Proceedings B.

Open Access

You are invited to opt for Open Access, making your freely available to all as soon as it is ready for publication under a CCBY licence. Our article processing charge for Open Access is £1700. Corresponding authors from member institutions (<http://royalsocietypublishing.org/site/librarians/allmembers.xhtml>) receive a 25% discount to these charges. For more information please visit <http://royalsocietypublishing.org/open-access>.

Paper charges

Sincerely,

Appendix A

Review: McMahon "Earth's earliest and deepest purported microfossils may be iron-mineralized chemical gardens".

This paper reports chemical garden experiments that produce Fe-mineralized filaments that show morphological similarities to structures found in the rock record of argued biogenicity. I find this an appealing and attractive idea that raises the profile of an abiotic process that could generate pseudofossils. In the context of the oldest candidate Fe-oxidising microfossils (Dodd et al. 2017) I think this work is particularly relevant, and I share the author's concerns about the biogenicity of these purported microfossils. I therefore think the current study is worthy of publication. I have several concerns about the current manuscript however, which largely stem from the short format and relatively small number of experiments that were conducted (2 Figures only).

- 1) Previous work by Garcia-Ruiz and colleagues is mentioned on silica-carbonate chemical gardens but not given sufficient credit. These authors have worked extensively to relate their chemical garden experiments to real geological environments, providing a stronger analogue than the current study. At least two further papers that deserve mention:
 - Archean N Pole Dome microfossil analogue, where organic C was condensed onto the microfossil mimics in a hydrothermal environment (Garcia-Ruiz et al. 2003 Science)
 - East African Rift valley studies, where the chemical garden experiments were compared directly to changing depositional environments found in these alkaline lakes. (e.g. Garcia-Ruiz et al. 2012 SEPM).
- 2) The authors focus on Fe-filament morphology in isolation, and do not discuss much about how filaments are arranged/distributed in nature versus in their experiments, which can provide important biogenicity constraints. In a study by my PhD student under final revision for Geobiology (Johannessen et al. attached for the author) we have studied a hydrothermal deposit in which, biogenic and abiogenic Fe-filaments occur alongside one another. Here we were able to systematically develop criteria to tell the two apart, using their size, branching patterns, internal microstructure, and growth/organisation. I would strongly advise the authors to give this some thought compared to what they find in their petri-dish experiments. Particularly considering the density of filaments and their arrangement/orientation as this can be readily observed in thin section. Quantitative data exist in the literature for natural populations of Fe-filaments that could be compared to the chemical garden experiments in this study (e.g. Krepski et al. 2013 Geobiology).
- 3) Twisted/helical shaped filaments are very diagnostic of certain groups of Fe-oxidising bacteria (*Galionella*), and in my opinion are absent from the oldest fossil claims. Were twisted filaments ever produced by your chemical garden experiments? Apparently not? Discuss further.
- 4) A quantitative morphospace of the filaments produced by your experiments, or at least size distribution plots would significantly strengthen this work and aid comparisons to real geological environments. As you note this has recently been accomplished for silica-witherite

biomorphs (Rouillard et al. 2018). Can you provide such data, at least on size distribution? Please add to the supplementary info if space does not allow.

- 5) Regarding the comparison to the oldest claims for fossil fungi, I find two features in your experiments interesting: apparent anastomosis (bridging filaments) and swellings along the filaments. However, I disagree about the mineral paragenesis of the Ongeluk filaments (Bengtson et al. 2017). These do comprise Fe-rich chlorite (chamosite-chlinoclore) which is common in metavolcanic rocks, but this does not necessarily mean that the filaments were originally composed of clays and Fe-oxyhydroxides and that the latter possibly formed through chemical garden type processes. I find it more likely that the filaments were mineralized by zeolites and/or clays on the subseafloor, and there is not strong evidence for abundant primary Fe-oxyhydroxides as traces of Fe-oxides are not found in/on these filaments. I think your comparison does not hold here, please discuss and/or defend further.

Reference:

Johannessen et al. (2019). *On the biogenicity of Fe-oxyhydroxide filaments in silicified low-temperature hydrothermal deposits: implications for the identification of Fe-oxidizing bacteria in the rock record*. *Geobiology* (please do not disseminate this paper as it is pending acceptance of final minor revisions in the journal, I will send you the finalised proofs once received, but in the meantime I hope you find this helpful).

In summary, I would like to see this study published after **moderate revisions** and consideration of the points above, **especially more info on the distribution and organisation of the experimental filaments and their size distribution**. I would be happy to look at a revised version of this ms.

Nicola McLoughlin

Rhodes University

August 2018

Appendix B

Please find below my response to the referees. I sincerely thank them for their time and efforts, which have resulted in a large number of revisions to my manuscript. I have indicated my revisions with **line numbers in red**; the line numbers refer to the revised document.

Each reviewer comment has been given a number to aid cross-referencing in my responses.

REVIEWER 1

1. This manuscript addresses a crucial question in geobiology: the difficulties of distinguishing between non-biogenic and biogenic objects of simple morphologies.

I agree with the reviewer about the importance of this question.

2. Life-like morphologies can be produced by physicochemical processes, and one well-known example is the “chemical gardens” often sold as toys, which produce intricate plant-like patterns by the addition of metal salts to a silicate solution. These patterns are the result of complex interplays between osmosis and precipitation. The relevance for the interpretation of putative microbial fossils has been studied in particular by the research group of J.M. García-Ruiz, who has demonstrated various life-like shapes, including “leaves” and “cauliflowers”, as well as filaments of different dimensions that may be straight, curved, simple, branching, but usually helicoidal.

“Chemical garden” is quite a broad term, but I would stress that classical chemical gardens (which García-Ruiz refers to as “metal silicate hydrate” structures) have not previously been compared to putative fossils in any detail, although they have sometimes been mentioned briefly in this context. The famous work of García-Ruiz concerns silica-carbonate biomorphs which do not much resemble the “fossils” at issue in this paper either in morphology or in composition, as I explain in the manuscript (now in slightly more detail) on **lines 88–92.**

3. Although these phenomena are known and typically cited in studies of ancient putative microbial fossils, interpretations of the fossils are frequently controversial on account of the biogenicity problem. Also even where there is geochemical evidence from organic carbon in support, this is often not enough, since evidence of biological activity does not necessarily mean that the studied morphologies are biologically generated.

I agree with the reviewer about the limited significance of organic carbon as a biosignature, a point also raised by reviewer 2. I have now addressed it by pointing out that organic carbon does not demonstrate biogenicity, on **lines 88–90 and again on **lines 370–372**.**

4. McMahon presents experiments producing ferric-encrusted filaments by

seeding ferrous sulfate granules (<63 µm) in silicate or carbonate solutions, a version of chemical gardens. The growth space was vertically restricted by the use of Petri dishes. The obtained structures resemble filamentous tubular biomorphs previously reported from chemical gardens by, e.g., Haudin et al. 2015 (PCCP 17:12804–12811) and Brau et al. 2018 (PCCP 20:784–793). These authors used a vertically confined space of 0.5 mm for their experiments. McMahon’s structures are more life-like in that they are more frequently branching; otherwise there is little to distinguish them morphologically. Similarly, the experimentally produced minute tubes reported by Makki et al. 2009 (Angewandte Chemie International Edition 48:8752–8756) are similar in dimensions and morphology, although they do not show branching. In particular, Makki et al. show how fast growth produces smooth and comparatively straight tubes, whereas slower growth results in tubes that appear grainy and segmented. None of the references above are in McMahon’s manuscript.

I thank the reviewer for taking the time to provide these helpful references, which I agree should be cited. I now draw readers’ attention to the papers by Haudin et al. and Brau et al on lines 216-221, and I discuss the result of Makki et al. on lines 388-390 as part of a new discussion of the potential importance of filament microtexture/structure (lines 382-396). The present paper does not aim primarily to lengthen the list of morphologies reported from classical chemical gardens, but to *show the extent of their chemical and morphological resemblance to putative fossils*, an important fact not well known to palaeontologists (e.g., none of palaeontological literature I cite discusses this possibility).

5. McMahon’s claim for anastomoses, a salient feature of fungal mycelia and of the Palaeoproterozoic Ongeluk fungus-like structures discussed in the manuscript, seems poorly founded. The alleged anastomoses are said to occur “at any angle” but to be rare, and the only figured example (Fig. 1H) is singularly unconvincing, as it shows only a minor detachment from and reunification with the same filament. This might not even be a growth feature, but an instance of parallel fibres being somewhat teased apart. A true anastomosis implies a branch seeking out a filament other than that which it originates from, and merging with it, and this is not shown in Fig. 1H.

I agree with the reviewer’s point that the illustration showed only a minor anastomosis. I have replaced this panel with a better one.

6. Significantly, the filaments reported by Haudin et al. and Brau et al. (refs above) showed a clear self-avoidance, seemingly precluding the formation of anastomoses.

I agree that this is significant and interesting, and now refer to this result on lines 218-221. In my experiments, anastomosis was indeed rare, but filaments sometimes met and grew along each other as if stuck together but unable to converge into a single filament, a behavior apparently not seen in these other studies. I have added mention of this

behavior in the Results on lines 158-159 and in the Discussion on lines 220-221.

7. There are other features of the Ongeluk structures that remain unexplained by the chemical gardens. In the Ongeluk vesicles, the filaments form a jumbled mat adhering to the rock walls which then sends out filaments into the voids. This behaviour is neither observed nor readily explained in a chemical model.

The Ongeluk filaments are frequently entangled in a way that indicates original flexibility, and they tend to form broom-like aggregates. This habitus has still to be demonstrated in chemical-garden experiments.

I agree that these features are not seen in my experiments, and I have added some discussion of this fact on lines 323-333. However, I disagree with the reviewer that these features are not readily explained by chemical gardens. The “mat” certainly looks as though it could have formed by the convergence and mineralization of the membrane material that appears in the early stage of chemical gardening. I now raise this possibility on lines 330-333. The tubes produced by chemical gardens are highly flexible in the early, gelatinous stage of growth, before they mineralize and become brittle; I now point out this important fact on lines 130-132 and line 328. Fundamentally though, I take the reviewer’s point, and I conclude this new discussion text by pointing out that further work is necessary to test my proposals (line 333-335).

8. The Archean/Hadean Nuvvuagittuq structures, like the Ongeluk ones, lack chemical evidence of an originally organic composition, and they are morphologically simpler than the latter. McMahon’s discussion of biogenicity in the Nuvvuagittuq is more to the point, and the arguments temper the cautiously positive reception these structures have received in the literature, e.g., Whitehouse et al. 2019 (Precambrian Research 323:70–81).

I thank the reviewer for this supportive statement.

9. The experimental results in the manuscript add to the known variety of biomorphs produced by abiotic processes. Eventually the paper should be published, but it must then include a fuller treatment of the literature (cf. the references given above that present results similar to the ones in the manuscript), and the important claim about anastomosing filaments must be backed up by documentation or dropped altogether. Arguments for an abiogenic origin of the Ongeluk filaments need to be made stronger or dropped.

I thank the reviewer for this broadly supportive comment and restatement of the main points of their extremely helpful and incisive review, which are addressed individually by the foregoing remarks. I have opted to make my discussion of the Ongeluk filaments more circumspect and nuanced rather than to put the case with unwarranted confidence or drop it altogether; readers can assess the case themselves. This seems the most helpful way I can contribute to this

debate, and I hope it will be acceptable to the reviewer.

10. A stylistic point: A single author should not refer to himself as “we”.

I have revised all uses of the first person plural to the first person singular. I will of course defer ultimately to the editors on all matters of style.

REVIEWER 2

1. This paper reports chemical garden experiments that produce Fe-mineralized filaments that show morphological similarities to structures found in the rock record of argued biogenicity. I find this an appealing and attractive idea that raises the profile of an abiotic processes that could generate pseudofossils. In the context of the oldest candidate Fe-oxidising microfossils (Dodd et al. 2017) I think this work is particularly relevant, and I share the author’s concerns about the biogenicity of these purported microfossils. I therefore think the current study is worthy of publication.

I thank the reviewer for these supportive remarks.

2. I have several concerns about the current manuscript however, which largely stem from the short format and relatively small number of experiments that were conducted (2 Figures only).

Previous work by Garcia-Ruiz and colleagues is mentioned on silica-carbonate chemical gardens but not given sufficient credit. These authors have worked extensively to relate their chemical garden experiments to real geological environments, providing a stronger analogue than the current study. At least two further papers that deserve mention:

- Archean N Pole Dome microfossil analogue, where organic C was condensed onto the microfossil mimics in a hydrothermal environment (Garcia-Ruiz et al. 2003 *Science*)
- East African Rift valley studies, where the chemical garden experiments were compared directly to changing depositional environments found in these alkaline lakes. (e.g. Garcia-Ruiz et al. 2012 *SEPM*).

I thank the reviewer for highlighting these papers by Garcia-Ruiz, and I agree that they merit additional discussion in the present study, notwithstanding the space constraints. The possibility of organic condensation onto inorganic biomorphs is now pointed out on **lines 88–90 and line 371, with reference to Garcia-Ruiz et al. 2003, *Science*. The *SEPM* paper (published in 2000 and included in a 2012 collection) is highly significant in pointing out that alkaline, silica-rich waters occur in a natural setting that I had not previously mentioned; I now do so on **lines 346-348**, with a citation to this paper.**

3. The authors focus on Fe-filament morphology in isolation, and do not discuss much about how filaments are arranged/distributed in nature versus in their experiments, which can provide important biogenicity constraints.

I agree that I have not shown experimentally whether naturally occurring chemical garden filament populations could be arranged in space in the same way as the iron mineral filament assemblages under debate. As reviewer 1 pointed out, this seems particularly problematic in respect of the Ongeluk filaments, and I now discuss this problem with new text on **lines 323-333, alluding to the fact that filament organisation in space critically depends upon the arrangement of the seed material (or its precursor), the flow regime in which chemical gardens grow, and the trigger for the onset of chemical gardening. Addressing this experimentally would require a new approach, perhaps involving the percolation of the silica solution (possibly at high P/T) through a porous medium pre-dosed with a coating of seed material. My main point in this paper is that even with a much more straightforward approach, one can tick off the “morphological criteria” for biogenicity. However, I do agree with the reviewer’s point and think this is an important direction for future research. I have therefore made this fully explicit on **lines 333-335** and on **line 409**.**

4. In a study by my PhD student under final revision for Geobiology (Johannessen et al. attached for the author) we have studied a hydrothermal deposit in which, biogenic and abiogenic Fe-filaments occur alongside one another. Here we were able to systematically develop criteria to tell the two apart, using their size, branching patterns, internal microstructure, and growth/organisation. I would strongly advise the authors to give this some thought compared to what they find in their petri-dish experiments.

I thank the reviewer for directing me to this excellent new paper, which I now cite in numerous places for its insights:

- On **line 76** in relation to diffusion-limited dendritic growth
- On **line 155** in relation to the supposed difficulty of bifurcation in abiotic systems
- On **line 227** in relation to helical forms
- On **line 238** as another example of oxyhydroxide filaments
- On **lines 250 and 341** as a published example of possible chemical gardens occurring in nature
- On **lines 375, 378 and 392** in relation to possible differences between biotic and abiotic filaments raised by the paper.

Nevertheless, I do not think the biogenicity criteria enumerated by Johannessen et al. are fully generalizable (although they may be correct in respect of the particular Arctic Mid-Ocean Ridge material investigated). My results and the sources I cite lead me to suspect that chemical garden filaments can reproduce essentially all the features in the left-hand column of Johannessen et al.’s Figure 9, where they are labelled as biosignatures.

5. Particularly considering the density of filaments and their arrangement/orientation as this can be readily observed in thin section. Quantitative data exist in the literature for natural populations of Fe-filaments that could be compared to the chemical garden experiments in this study (e.g. Krepeski et al. 2013 *Geobiology*).

I totally agree with the suggestion to include quantitative data for comparison with natural populations. I have accordingly produced a new **supplementary file containing the diameters of 200 filaments produced from sieved ferrous sulfate grains, as measured using ImageJ on the accompanying photomicrographs; I have summarised the results on **lines 135–139**. (The diameter is of course much a more useful and meaningful parameter than the length, which is rarely fully apparent in thin section). I have also provided some accompanying discussion of the results on **lines 255–260**. As I state there, the results are suggestively similar to previously published measurements of filaments in geological materials. I have resisted the temptation to conduct a detailed quantitative comparison with the fossil record because I think this should be reserved for experiments conducted under more realistic conditions, e.g., filament growth from sulphide linings in basalt without artificially sieving out grains above 63 microns.**

In relation to arrangement and orientation, please see my response to this reviewer's comment 3.

6. Twisted/helical shaped filaments are very diagnostic of certain groups of Fe-oxidising bacteria (*Galionella*), and in my opinion are absent from the oldest fossil claims. Were twisted filaments ever produced by your chemical garden experiments? Apparently not? Discuss further.

I agree with the reviewer that this morphology may be significant. I have now pointed out on **lines 225–229 that helical forms did not occur in my experiments, but were reported by Leduc (1911). Twisted forms can of course also occur in Garcia-Ruiz-type silica-carbonate biomorphs.**

7. A quantitative morphospace of the filaments produced by your experiments, or at least size distribution plots would significantly strengthen this work and aid comparisons to real geological environments. As you note this has recently been accomplished for silica-witherite biomorphs (Rouillard et al. 2018). Can you provide such data, at least on size distribution? Please add to the supplementary info if space does not allow.

I agree — see response to comment 5.

8. Regarding the comparison to the oldest claims for fossil fungi, I find two features in your experiments interesting: apparent anastomosis (bridging filaments) and swellings along the filaments. However, I disagree about the mineral paragenesis of the Ongeluk filaments (Bengtson et al. 2017). These do comprise Fe-rich chlorite (chamosite-chlinoclore) which is common in metavolcanic rocks, but this does not necessarily mean that the filaments were originally composed of clays and Fe-oxyhydroxides and that the latter

possibly formed through chemical garden type processes. I find it more likely that the filaments were mineralized by zeolites and/or clays on the seafloor, and there is not strong evidence for abundant primary Fe-oxyhydroxides as traces of Fe-oxides are not found in/on these filaments. I think your comparison does not hold here, please discuss and/or defend further.

I agree that the paragenesis of the iron-rich chlorite is open to several interpretations, so I have expanded the discussion as requested (lines 307-315). Fe oxyhydroxides and silica can react to form iron-rich smectite in marine hydrothermal settings (Dekov et al., *Chemical Geology*, 2007) and perhaps there was simply no Fe oxide left. It is surely relevant that chlorite constitutes the greenish, dendritic, filamentous “moss” in the moss agates found in basalt amygdales, a dubiofossil which ranges from dendritic, obviously abiogenic forms to other forms that have been interpreted by some as biogenic. However, I do not think there is really clear evidence one way or the other about the origin of the Ongeluk chlorite. The revised discussion reflects this.

9. In summary, I would like to see this study published after moderate revisions and consideration of the points above, especially more info on the distribution and organisation of the experimental filaments and their size distribution. I would be happy to look at a revised version of this ms.

I thank the reviewer for their close attention to my manuscript and their extremely helpful critical feedback, summarised in this comment.

REVIEWER 3

1. This work is very important, since abiotic processes may form chemical and morphological features similar to biological processes. This fact may result in misinterpretation of fossils and biosignatures if care is not taken. So, this work is therefore an important contribution to the research fields on early life on earth and fossilized microorganisms. The paper is well structured and concise. I recommend accept with a few minor revisions.

I thank the reviewer for these supportive remarks.

2. It would be interesting to see some more in depth comparative discussion on the papers by Glaab et al. 2016 or Opel et al. 2019, where they use iron salts and lower pH. The paper by Glaab is mentioned but a comparison with this work would be very interesting, I think.

I thank the reviewer for returning my attention to these interesting papers. Glaab et al. use compressed salt pellets to produce large tubes from ferrous and ferric chloride so that they can probe the chemical gradients across the membranes. The overall morphology is not relevant to the present work, but the microtextural details in their Fig. 4 (SEM micrographs) are highly interesting and I have now cited this

aspect of the study on line 391. Work by Opel et al. focuses on silica carbonate biomorphs which show interesting but different morphologies as discussed on lines 80-95 and 218-222; I did not feel an additional citation to Opel et al's work was needed.

3. Are there any natural environments today where biomorphs have been found? The authors talk about plausibility of chemical garden growth in nature but do not give actual examples of places where any of these gardens are found? Except for fossilized structures? Are there any places where chemical gardens are forming these structures today? It would be helpful with some concrete examples on where to look. There are examples such as a) deep, isolated groundwaters and b) silica-rich alkaline springs but no evidence on actual chemical gardens have been found there yet. If not, why? Are there other places to look for those structures? What would such sampling require? The authors discuss briefly on these matters but it would be interesting to know more and a bit more in depth.

The reviewer raises an interesting question, which has been discussed in detail by Barge et al. (2012), who refer to modern and ancient hydrothermal vents and mounds as large, complex, chemical gardens, and to brinicles as an example of a similar process based on different chemistry. If I am right, many of the iron-mineral filaments in the rock record and growing at modern hydrothermal systems are also smaller-scale chemical gardens. The new Johannessen et al. paper mentioned by reviewer 2 also makes the novel suggestion that silica tubes in the same settings could be chemical gardens, and I have cited it to that effect on lines 250 and 341. I have revised lines 340-341 to read, "most notably forming complex structures at deep-sea hydrothermal vents (e.g., 28; see also ref. 32 for a discussion of chemical gardens in nature)". If chemical gardens are actually forming in nature today, it would be difficult to show that they were not biologically mediated in some way given the ubiquity of microorganisms.

4. What kind of further experiments would be needed to distinguish between fossilized abiotic and biologic iron oxide tubes? Tagged samples? Are there examples of similar experiments that of the authors of this manuscript, but with added microorganisms? If so, are there differences in the morphology between abiotically and biologically formed iron oxide tubes?

These important questions together with the comments of the other reviewers have prompted me to add a somewhat speculative new section at the end of the Discussion entitled *Discriminating chemical gardens and iron-mineralized fossil microbes*. I am not aware of any published experiments combining chemical garden and biotic iron oxide mineralization but this would be very worthwhile (though clearly beyond my scope in this paper). There has also been minimal experimental work on the iron encrustation of filamentous fungi. I have added these specific recommendations for future work in revisions on lines 366-396.